# Effects of Short-Term Nitrogen Addition on Soil Fungal Community Increase with Nitrogen Addition Rate in an Alpine Steppe at the Source of Brahmaputra

**DOI:** 10.3390/microorganisms11081880

**Published:** 2023-07-25

**Authors:** Shaolin Huang, Chengqun Yu, Gang Fu, Wei Sun, Shaowei Li, Fusong Han, Jianyu Xiao

**Affiliations:** 1Lhasa Plateau Ecosystem Research Station, Key Laboratory of Ecosystem Network Observation and Modeling, Institute of Geographic Sciences and Natural Resources Research, Chinese Academy of Sciences, Beijing 100101, China; 2University of Chinese Academy of Sciences, Beijing 100049, China

**Keywords:** nitrogen deposition, soil fungi, community structure, alpine steppe, Brahmaputra

## Abstract

The soil fungal community plays a crucial role in terrestrial decomposition and biogeochemical cycles. However, the responses of the soil fungal community to short-term nitrogen addition and its related dominant drivers still remain unclear. To address this gap, we conducted an experiment to explore how different levels of nitrogen addition (five levels: 0, 2.5, 5, 10, and 20 g N m^−2^ y^−1^) affected the soil fungal community in an alpine steppe at the source of Brahmaputra. Results showed that the reduced magnitudes of soil fungal species and phylogenetic α-diversity increased with the increasing nitrogen addition rate. Nitrogen addition significantly changed the community composition of species, and the dissimilarity of the soil fungal community increased with the increasing nitrogen addition rate, with a greater dissimilarity observed in the superficial soil (0–10 cm) compared to the subsurface soil (10–20 cm). Increases in the soil nitrogen availability were found to be the predominant factor in controlling the changes in the soil fungal community with the nitrogen addition gradient. Therefore, short-term nitrogen addition can still cause obvious changes in the soil fungal community in the alpine grassland at the source of Brahmaputra. We should not underestimate the potential influence of future nitrogen deposition on the soil fungal community in the high-altitude grassland of the Qinghai–Tibet Plateau. Adverse effects on the soil fungal community should be carefully considered when nitrogen fertilizer is used for ecosystem restoration of the alpine grassland of the Qinghai–Tibet Plateau.

## 1. Introduction

With dramatically rising nitrogen (N) deposition worldwide during the last hundred years, studies on the effects of N deposition/addition on terrestrial ecosystems are increasing [1,2,3,4,5]. Preceding studies have concentrated on the responses of plant variables (e.g., productivity, α-diversity, community composition, and root trait), soil conventional physicochemical variables (e.g., pH, soil organic carbon, C:N, net N-mineralization, and base cations), soil microbial community (e.g., diversity, community composition, and biomass), and soil carbon storage to experimental N addition or atmospheric N deposition [5,6,7,8,9,10,11,12,13]. As is well known, the belowground soil microbial community is a crucial determining factor of their ecological functions [14,15]. Soil fungi are a diverse group of creatures, which play vital roles in governing terrestrial decomposition and biogeochemical cycles [16]. While some previous research has reported the response of the soil fungal community to nitrogen deposition [17,18,19], current studies have explored this topic far less than that of the soil bacterial community [20]. In addition, previous studies still have to address some issues that require further research. First, the response of the soil fungal α-diversity and community composition to short-term nitrogen addition gradients remains controversial [20,21,22,23,24], perhaps due to a range of nitrogen addition rates, as well as climate and plant conditions. Second, the predominant factor in controlling the changes in the soil fungal diversity following nitrogen addition is controversial [25,26], which may be related to the duration. Third, while previous research has examined the environmental factors which contribute to microbial community differences [27], the stochastic processes of community assembly also play an important role in microbial diversity [28], yet this has not been integrated with community assembly of soil fungi under nitrogen addition conditions [5,13,23]. Fourth, the soil fungal diversity contains species α-diversity and β-diversity, and phylogenetic α-diversity and β-diversity, which can deduce community assembly mechanisms [29]. However, few studies have focused on the response of phylogenetic α-diversity and β-diversity to nitrogen addition. Therefore, further experimental nitrogen addition studies are worthwhile to reveal the response of the soil fungal community to short-term nitrogen deposition and related influences, which can help us in predicting the dynamic response of the soil microbial community to nitrogen deposition.

As the highest plateau in the world, the high altitude and low temperature of the Tibetan Plateau result in a relatively fragile alpine ecosystem, which is more sensitive to global change than other regions [30]. On the one hand, the Tibetan Plateau has also experienced elevated N deposition in the last few decades [31]. On the other hand, alpine grasslands—one of the important ecosystems on the Qinghai–Tibet Plateau—have experienced degradation in recent decades [32], and fertilization is one of the important methods to recover degraded grasslands [33]. With the development of high-throughput sequencing technology, more and more studies have been conducted on the response of the soil fungal community to trial nitrogen addition in high-altitude grassland in the northeast of the Qinghai–Tibet Plateau [22,34,35]. However, many of these studies have focused on surface (0–10 cm) soil [22,35] rather than subsurface (10–20 cm) soil. In addition, no research has reported the influences of nitrogen addition on the soil fungal community in the high-altitude steppe in the west of the Tibetan Plateau. Considering its relatively greater magnitude of arid conditions, the alpine steppe in the west of the Tibetan Plateau may be more sensitive and vulnerable than other regions of the Tibetan Plateau [13]. Therefore, more experiments are required to evaluate the influences of nitrogen deposition on the soil fungal community in high-altitude grasslands on the Qinghai–Tibet Plateau. This is of great significance for predicting the influences of nitrogen deposition on the terrestrial ecosystem and for improving the management of degraded grassland.

The Brahmaputra River, the main river in the Himalayan Mountain range in Tibet, plays a crucial role as an ecological security barrier domestically and internationally. It connects China, India, and Bangladesh, ultimately merging into the Bay of Bengal [36]. Understanding the impact of nitrogen addition on the alpine steppe is essential for the Brahmaputra River basin. In this research, we explored the responses of soil fungal species and phylogenetic community structure (α-diversity and community composition) to short-term (<1 year) nitrogen addition based on an outdoor nitrogen addition experimentation with five treatments (i.e., N0, N2.5, N5, N10, and N20) in an alpine steppe at the source of Brahmaputra. We aimed to explore (1) how the nitrogen addition rate can influence the responses of the soil fungal α-diversity and community composition to short-term (<1 year) nitrogen addition in both surface (0–10 cm) and subsurface (10–20 cm) soils, and (2) what the predominant factor is in controlling the changes in the soil fungal community to short-term nitrogen addition with the N addition gradient in the alpine steppe at the source of Brahmaputra.

## 2. Materials and Methods

### 2.1. Study Site and Experimental Design

The study site was situated in an alpine steppe of Zhongba County (29°37′ N, 82°21′ E, 4763 m), Tibet Autonomous Region, China. The mean yearly temperature was about 5.1 °C, and the mean yearly precipitation was about 320 mm. Before the experiment, we detected the vegetation conditions and soil physicochemical properties of the study site. The dominant species in this steppe was *Potentilla bifurca*, accompanied by *Microula sikkimensis*, etc. Plant roots were mainly distributed at the 0–20 cm depth. The soil organic carbon and pH values were 0.57–0.81% and 8.27–8.47, respectively [37].

The detailed information about nitrogen adding had been described by Huang et. al [37] and is briefly summarized here. In August 2021, a homogenous steppe was selected for the nitrogen fertilization experiment using a one-way factor experimental design. The field experiment with five N addition rates (0, 2.5, 5, 10, and 20 g N m^−2^ y^−1^ of urea) was conducted, which were characterized by N0 (control), N2.5, N5, N10, and N20, respectively. Each treatment had five replicates, and a total of twenty-five 5 m × 5 m plots were arranged founded on a stochastic design. The distance between the two adjacent plots was about 3 m.

### 2.2. Vegetation Community Examination, Soil Sampling, and Analysis

In September 2021, we recorded the total coverage existing in a 0.5 m × 0.5 m quadrat in the middle of every plot, and the number of plant species, height, and coverage of every plant species [38]. After plant community investigation, we used an auger of 3.8 cm in diameter to collect soil at a depth of 0–10 cm and 10–20 cm from every quadrat. Five soil cores were collected and then mixed to take a composite soil sample. A 2.0 mm sieve was used to remove stones and roots in the compound soil samples. Some of the soil samples were used to measure the soil fungal OTUs, and some were used to measure the soil physicochemical properties, including pH, soil moisture (SM), ammonium nitrogen (NH_4_^+^-N), nitrate nitrogen (NO_3_^−^-N), available phosphorus (AP), soil organic carbon (SOC), total phosphorus (TP), total nitrogen (TN), and soil enzymes, including sucrase (SC), cellulase (CL), urease (UA), catalase (CT), and alkaline phosphatase (ALP). The rates of SOC to TN (C:N), SOC to TP (C:P), TN to TP (N:P), NH_4_^+^-N to NO_3_^−^-N (NH_4_^+^-N:NO_3_^−^-N), and the sum of NH_4_^+^-N and NO_3_^−^-N to available phosphorus (AN:AP) were then calculated.

We described the high-throughput sequencing procedures in detail in the Appendix A. Analysis revealed 2267 OTUs in the surface soil and 2470 OTUs in the subsurface soil of the soil fungal community across the 50 samples.

### 2.3. Statistical Analysis

The response ratio was calculated to obtain the effect size of nitrogen addition for a particular variable [7,39].
(1)$$R=\frac{Ni}{N0}$$
where *Ni* was the nitrogen addition rate and *N*0 was the control treatment.

*R*_OTUs_: effect size of nitrogen addition on operational taxonomic units of soil fungal community; *R*_ACE_: effect size of nitrogen addition on ACE index of soil fungal community; *R*_Chao1_: effect size of nitrogen addition on Chao1 index of soil fungal community; *R*_Shannon_: effect size of nitrogen addition on Shannon index of soil fungal community; *R*_Simpson_: effect size of nitrogen addition on Simpson index of soil fungal community; *R*_PD_: effect size of nitrogen addition on phylogenetic diversity of soil fungal community; *R*_MPD_: effect size of nitrogen addition on mean pairwise distance of soil fungal community; *R*_MNTD_: effect size of nitrogen addition on mean nearest taxon distance of soil fungal community; *R*_NRI_: effect size of nitrogen addition on net relatedness index of soil fungal community; *R*_NTI_: effect size of nitrogen addition on nearest taxon index of soil fungal community.

*R*_Shannonplant_: effect size of nitrogen addition on Shannon index of plant community; *R*_Simpsonplant_: effect size of nitrogen addition on Simpson index of plant community; *R*_SRplant_: effect size of nitrogen addition on species richness of plant community; *R*_Pielouplant_: effect size of nitrogen addition on Pielou index of plant community; *R*_SM_: effect size of nitrogen addition on soil moisture; *R*_AP_: effect size of nitrogen addition on available phosphorus; *R*_NH4+-N_: effect size of nitrogen addition on ammonium nitrogen; *R*_NO3−-N_: effect size of nitrogen addition on nitrate nitrogen; *R*_SOC_: effect size of nitrogen addition on soil organic carbon; *R*_TP_: effect size of nitrogen addition on total phosphorus; *R*_TN_: effect size of nitrogen addition on total nitrogen; *R*_C:N_: effect size of nitrogen addition on ratio of carbon to nitrogen; *R*_C:P_: effect size of nitrogen addition on ratio of carbon to phosphorus; *R*_N:P_: effect size of nitrogen addition on ratio of nitrogen to phosphorus; *R*_NH4+-N:NO3−-N_: effect size of nitrogen addition on ratio of ammonium nitrogen to nitrate nitrogen; *R*_AN:AP_: effect size of nitrogen addition on ratio of available nitrogen to available phosphorus; *R*_SC_: effect size of nitrogen addition on soil sucrase; *R*_CL_: effect size of nitrogen addition on soil cellulase; *R*_UA_: effect size of nitrogen addition on soil urease; *R*_CT_: effect size of nitrogen addition on soil catalase; and *R*_ALP_: effect size of nitrogen addition on soil alkaline phosphatase.

We used the Microeco package to obtain the soil fungal species α-diversity (OTUs, ACE, Chao1, Shannon, and Simpson) and β-diversity [40]. ACE and Chao1 were proposed by Chao to estimate the number of OTUs in a community [41]. We used the Picante package to calculate the soil fungal phylogenetic α-diversity (PD, MPD, MNTD, NRI, and NTI) and β-diversity. We used the Microeco package to carry out a differential abundance test for soil fungal species based on the LEfSe method and then used Duncan multiple comparison to calculate the differential species of soil fungi. We used the Random Forest package to calculate the relative contribution to soil fungal species and phylogenetic α-diversity of every relevant variable [42]. We used the ‘adonis2’ function to test the soil fungal species and phylogenetic community composition between any two treatments with the nitrogen addition gradient. We used the Hmisc package to calculate the Pearson correlations between soil fungal species and phylogenetic α-diversity and soil and plant variables. We used the Microeco package to calculate the Mantel test between species and phylogenetic community composition of soil fungi and environmental variables. We used the iCAMP package to calculate the community assembly processes of the soil fungal community [29,43,44]. Software used in this study include R 4.0.2, SPSS 25, and Origin 2023.

## 3. Results

### 3.1. Changes in Soil Fungal Species and Phylogenetic α-Diversity with the Nitrogen Addition Gradient and the Relation with Environmental Variables

The effect size of soil fungal species α-diversity, including OTUs, ACE, and Chao1 in surface soil and OTUs in subsurface soil, and soil fungal phylogenetic α-diversity, including PD, MPD, and NTI in surface soil and PD in subsurface soil, decreased with an increasing nitrogen addition rate (Figure 1a–d and Figure 2a–d). However, the effect size of soil fungal species α-diversity, including Shannon, Simpson in surface soil and ACE, Chao1, Shannon, and Simpson in subsurface soil, and soil fungal phylogenetic α-diversity, including MNTD and NRI in surface soil and MPD, NTI, MNTD, and NRI in subsurface soil, showed no relationships with the nitrogen addition rate (Figure 1e,f, Figure 2e,f, Appendix A and Appendix A).

The environmental variations included *R*_SM_, *R*_NO3−-N_, *R*_CL_, *R*_Shannonplant_, *R*_SC,_
*R*_TP_, etc., and they commonly explicated 61.58% variation of the *R*_OTUs_ in surface soil with the nitrogen addition gradient (Figure 3a). The environmental variations included *R*_NO3−-N_, *R*_CL_, *R*_CT_, *R*_Shannonplant_, *R*_Pielouplant_, *R*_NH4+-N:NO3−-N_, *R*_SM_, etc., and they commonly explicated 54.79% variation of the *R*_ACE_ in surface soil with the nitrogen addition gradient (Figure 3b). The environmental variations included *R*_CL_, *R*_Pielouplant_, *R*_SM_, *R*_Shannonplant_, *R*_NO3−-N_, etc., and they commonly explicated 52.83% variation of the *R*_Chao1_ in surface soil with the nitrogen addition gradient (Figure 3c). The environmental variations included *R*_TN_, *R*_C:N_, *R*_CL_, *R*_CT_, *R*_SC_, *R*_NO3−-N_, etc., and they commonly explicated 65.32% variation of the *R*_Shannon_ in surface soil with the nitrogen addition gradient (Appendix A). The environmental variations included *R*_SM_, *R*_CL_, *R*_ALP_, *R*_TN_, etc., and they commonly explicated 91.32% variation of the *R*_Simpson_ in surface soil with the nitrogen addition gradient (Appendix A). Further study found that *R*_SM_, *R*_NO3−-N_, etc., were dominant factors leading to significant changes in soil fungal species α-diversity, including *R*_OTUs_, *R*_ACE_, and *R*_Chao1_ in surface soil.

The environmental variations included *R*_CT_, *R*_pH_, *R*_NO3−-N_, *R*_SM_, *R*_SRplant_, *R*_C:N_, *R*_TP_, etc., and they commonly explicated 68.12% variation of the *R*_OTUs_ in subsurface soil with the N addition gradient (Figure 3d). The environmental variations included *R*_pH_, *R*_CL_, *R*_ALP_, *R*_CT_, *R*_NO3−-N_, etc., and they commonly explicated 75.12% variation of the *R*_ACE_ in subsurface soil with the N addition gradient (Figure 3e). The environmental variations included *R*_CT_, *R*_CL_, *R*_pH_, etc., and they commonly explicated 73% variation of the *R*_Chao1_ in subsurface soil with the N addition gradient; in particular, the predominant factor *R*_CT_ had the most relative contribution (Figure 3f). The environmental variations included *R*_UA_, *R*_AP_, *R*_pH_, *R*_NH4+-N:NO3−-N_, *R*_SC_, *R*_NH4+-N_, etc., and they commonly explicated 65.94% variation of the *R*_Shannon_ in subsurface soil with the N addition gradient (Appendix A). The environmental variations included *R*_AP_, *R*_UA_, *R*_CL_, *R*_pH_, *R*_SOC_, etc., and they commonly explicated 91.88% variation of the *R*_Simpson_ in subsurface soil with the N addition gradient (Appendix A). Further study still found that *R*_CT_, *R*_pH_, *R*_NO3−-N_, etc., were the leading factors resulting in significant change in soil fungal species α-diversity, including *R*_OTUs_ in subsurface soil.

The environmental variations included *R*_SM_, *R*_TP_, *R*_NO3−-N_, *R*_NH4+-N:NO3−-N_, *R*_CL_, *R*_CT,_ etc., and they commonly explicated 60.12% variation of the *R*_PD_ in surface soil with the nitrogen addition gradient (Figure 4a). The environmental variations included *R*_C:P_, *R*_TN_, *R*_CL_, *R*_NO3−-N_, etc., and they commonly explicated 74.54% variation of the *R*_MPD_ in surface soil with the nitrogen addition gradient (Figure 4b). The environmental variations included *R*_TN_, *R*_AP_, *R*_NH4+-N:NO3−-N_, *R*_CL_, *R*_C:N_, *R*_TP_, etc., and they commonly explicated 43.19% variation of the *R*_NTI_ in surface soil with the nitrogen addition gradient (Figure 4c). The environmental variations included *R*_Shannonplant_, *R*_CL_, *R*_AP_, *R*_SC_, *R*_C:P_, *R*_N:P_, *R*_pH_, etc., and they commonly explicated 42.68% variation of the *R*_MNTD_ in surface soil with the nitrogen addition gradient (Appendix A). The environmental variations included *R*_SOC_, *R*_C:P_, *R*_UA_, *R*_Shannonplant_, *R*_pH_, etc., and they commonly explicated 63.70% variation of the *R*_NRI_ in surface soil with the nitrogen addition gradient (Appendix A). Further research also found that *R*_SM_, *R*_NO3−-N_, *R*_NH4+-N:NO3−-N_, etc., were the dominant factors that caused significant changes in the soil fungal phylogenetic α-diversity, including *R*_PD_, *R*_MPD_, and *R*_NTI_ in surface soil.

The environmental variations included *R*_CT,_
*R*_pH_, *R*_NO3−-N_, *R*_TP_, *R*_C:N_, etc., and they commonly explicated 69.15% variation of the *R*_PD_ in subsurface soil with the nitrogen addition gradient (Figure 4d). The environmental variations included *R*_UA_, *R*_SC_, *R*_pH_, *R*_SOC_, etc., and they commonly explicated 68.32% variation of the *R*_MPD_ in subsurface soil with the nitrogen addition gradient (Figure 4e). The environmental variations included *R*_pH_, *R*_UA_, *R*_SOC_, *R*_C:P_, *R*_N:P_, etc., and they commonly explicated 49.70% variation of the *R*_NTI_ in subsurface soil with the nitrogen addition gradient (Figure 4f). The environmental variations included *R*_UA_, *R*_CT_, *R*_pH_, *R*_TP_, etc., and they commonly explicated 70.28% variation of the *R*_MNTD_ in subsurface soil with the nitrogen addition gradient (Appendix A). The environmental variations included *R*_pH_, *R*_TP_, *R*_ALP_, *R*_Pielouplant_, *R*_SM_, etc., and they commonly explicated 58.05% variation of the *R*_NRI_ in subsurface soil with the nitrogen addition gradient (Appendix A). Further study also found that *R*_CT,_
*R*_pH_, *R*_NO3−-N_, etc., were the dominant factors leading to significant changes in the soil fungal phylogenetic α-diversity, including *R*_PD_ in subsurface soil.

### 3.2. Changes in Soil Fungal Species and Phylogenetic Community Composition with the Nitrogen Addition Gradient and the Relation with Environmental Variables

The soil fungal species β-diversity between the control and nitrogen addition treatments significantly increased with the increasing nitrogen addition rate in surface soil (Figure 5a, *p* < 0.01) but not in subsurface soil (Figure 5b). For the soil fungal species community, the Adonis2 texts showed that nitrogen addition treatments significantly changed the community composition (*p* = 0.02) and N0 treatment was extremely divergent to the N20 treatment in surface soil, while not in subsurface soil (Table 1). There were 10 diverse clusters of soil fungal species community in surface soil whose LDA points were >2, including one class, two orders, two families, three genera, and two species (Appendix A). There were eight diverse clusters of soil fungal species community in subsurface soil whose LDA points were >2, including one phylum, two orders, four families, and one genus (Appendix A). Duncan multicomparison showed that o__Xylariales under control treatment was significantly higher than that under N20 treatment in surface soil (Appendix A). Duncan multicomparison also showed that g__Vishniacozyma under control treatment was significantly higher than that under others’ treatments in surface soil (Appendix A).

The soil fungal phylogenetic β-diversity showed no relationship with the nitrogen addition rate (Appendix A). For the soil fungal phylogenetic community, the Adonis2 texts showed that N0 treatment was extremely divergent to the N20 treatment in the surface soil but not in the subsurface soil (Table 1).

According to the results of the Mantel test, the soil fungal species community composition was significantly correlated with NO_3_^−^-N, NH_4_^+^-N, and SC in the surface soil. The soil fungal species community composition was significantly correlated with UA, SC, and TN in the subsurface soil (Table 2).

The soil fungal phylogenetic community composition was significantly correlated with SC and marginally correlated with SM, NO_3_^−^-N, NH_4_^+^-N, and AN:AP in the surface soil, respectively. The soil fungal phylogenetic community composition was marginally correlated with Shannon_plant_, TN, SC, and UA in the subsurface soil, respectively (Table 2).

### 3.3. Effect of Nitrogen Addition on Community Assembly of Soil Fungal Community

The change in the effect size of nitrogen addition on the ecological processes in both surface and subsurface soil is shown in Figure 6 and Appendix A. The ‘Dispersal Limitation’ process increased with the increasing nitrogen addition rate in surface soil (Figure 6a, *p* = 0.05), while others showed no relationships with the nitrogen addition rate (Figure 6b and Appendix A).

## 4. Discussion

### 4.1. Changes in Species and Phylogenetic α-Diversity with the Nitrogen Addition Gradient

We discovered that the reduced magnitude of soil fungal species and phylogenetic α-diversity enlarged with the increasing nitrogen addition rate, which may be due to several reasons. First, the soil nitrogen availability is an important factor influencing the species and phylogenetic α-diversity of the soil fungal community [45,46]. The effect size of nitrogen addition on soil NO_3_^−^-N enlarged with the increasing nitrogen addition rate (Appendix A) but was negatively interrelated with the species and phylogenetic α-diversity of the soil fungal community (Appendix A). Second, the pH is another main reason influencing the species and phylogenetic α-diversity of the soil fungal community [46,47,48,49]. The effect size of nitrogen addition on the soil pH was positively interrelated with the species and phylogenetic α-diversity of the soil fungal community (Appendix A). However, the effect size of the soil pH showed no relationship with the nitrogen addition rate (Appendix A), and no significant change in soil pH among the five nitrogen addition treatments was observed [37]. Third, the soil moisture is also a chief reason influencing the α-diversity of the soil fungal community [50,51]. The effect size of nitrogen addition on soil moisture enlarged with the increasing nitrogen addition rate (Appendix A) but was negatively interrelated with the α-diversity of the soil fungal community (Appendix A). Fourth, the plant α-diversity may also be related to the α-diversity of the soil fungal community [52,53], but there were no significant changes in the plant α-diversity along the nitrogen addition gradient [37]. Fifth, the soil fungal community α-diversity has no correlation with the soil enzyme activity [54], and no significant changes in SC, CL, UA, CT, and ALP were observed among the five nitrogen addition treatments [37]. Sixth, the relative contributions of AP, NH_4_^+^-N, SOC, TP, TN, C:N, C:P, N:P, and AN:AP to the α-diversity of the soil fungal community were negligible.

### 4.2. Changes in Species and Phylogenetic Community Composition with the Nitrogen Addition Gradient

In this study, we found that short-term (<1 year) nitrogen addition treatments and a high nitrogen addition rate (20 g N m^−2^ y^−1^) could significantly change the soil fungal community composition in surface soil in the alpine steppe at the source of Brahmaputra. This discovery was in accordance with several studies that proved that long-term (5–53 years) nitrogen addition (8–32 g N m^−2^ y^−1^) could significantly change the soil fungal community composition in grasslands or forests [5,18,23,55,56]. However, some studies reported that 2–3 years of nitrogen addition (10 g N m^−2^ y^−1^) did not significantly change the soil fungal community composition [22,34], which may be mainly due to the low nitrogen addition rate and short-term duration of nitrogen addition.

We found that the soil fungal β-diversity increased with the nitrogen addition rate in the surface soil, which may be due to several reasons. First, a previous study indicated that the available nitrogen is the key factor that influenced the soil fungal community composition [23]. In this study, the soil fungal community composition was significantly interrelated with NH_4_^+^-N and NO_3_^−^-N in surface soil (Table 2), and NH_4_^+^-N and NO_3_^−^-N significantly changed along the nitrogen addition gradient at the same time [37]. Second, the soil organic carbon is an important factor in determining the soil fungal community composition [57,58]. However, the soil organic carbon showed no significant changes with the increasing nitrogen addition rate [37]. Third, the total nitrogen, soil moisture, and soil enzyme activity have correlations with the soil fungal β-diversity [53,55,58], which did not significantly change across the nitrogen addition gradient [37]. Fourth, a previous study showed that the ‘Dispersal Limitation’ process results in relatively low similarity in the community [29]. In this study, the ‘Dispersal Limitation’ process increased with the nitrogen addition rate (Figure 6a).

### 4.3. Response Strength Is Stronger in Surface Soil than Subsurface Soil

In this research, the response strengths of the soil fungal community to nitrogen addition were different between soil depths, and the strength was stronger in surface soil than subsurface soil. There are a few reasons for this. First, it is well established that the soil fungal community differs between different soil depths due to different soil properties and community assembly processes in different soil depths [59,60]. Second, previous studies report that the enclosed microhabitat conditions in subsurface soil have been proven to be less affected by environmental changes than in surface soil [61,62]. Third, the soil fungal community is closely related to the available soil nitrogen, which varied in different soil depths due to nitrogen addition [63]. The predominant driving factors of soil NO_3_^−^-N had extremely significant differences between surface and subsurface soils.

## 5. Conclusions

Based on a short-term (<1 year) field nitrogen addition experiment in an alpine steppe at the source of Brahmaputra, we discovered that the negative effect of nitrogen addition on species and phylogenetic α-diversity of the soil fungal community and showed a decreased correlation with the nitrogen addition rate. Nitrogen addition significantly changed the species community composition of soil fungi, and the dissimilarity of soil fungal community caused by nitrogen addition increased with the increasing nitrogen addition rate. This phenomenon was stronger in surface soil than subsurface soil. Elevated nitrogen availability was the main driver of changes in soil fungal community. Therefore, the influences of nitrogen addition and the application of nitrogen fertilizer on the restoration of alpine steppes should be carefully reconsidered in the Brahmaputra River basin.

## Figures and Tables

**Figure 1 microorganisms-11-01880-f001:**
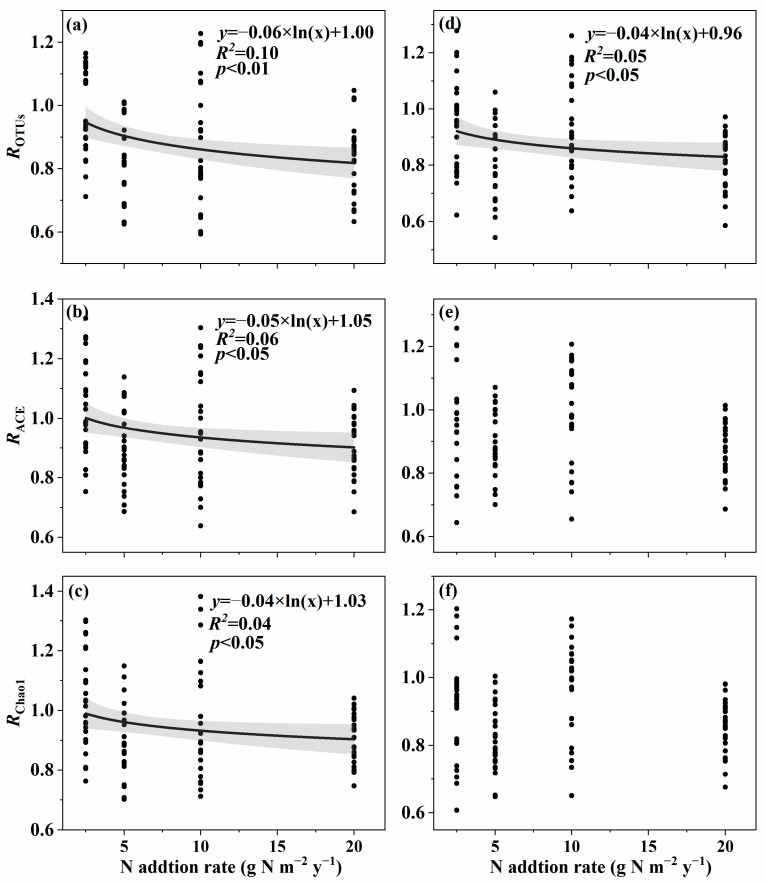
Relationships between nitrogen addition rate and the effect size of nitrogen addition on species α-diversity of soil fungi in an alpine steppe. Surface soil: (**a**) OTUs; (**b**) ACE; (**c**) Chao1. Subsurface soil: (**d**) OTUs; (**e**) ACE; (**f**) Chao1.

**Figure 2 microorganisms-11-01880-f002:**
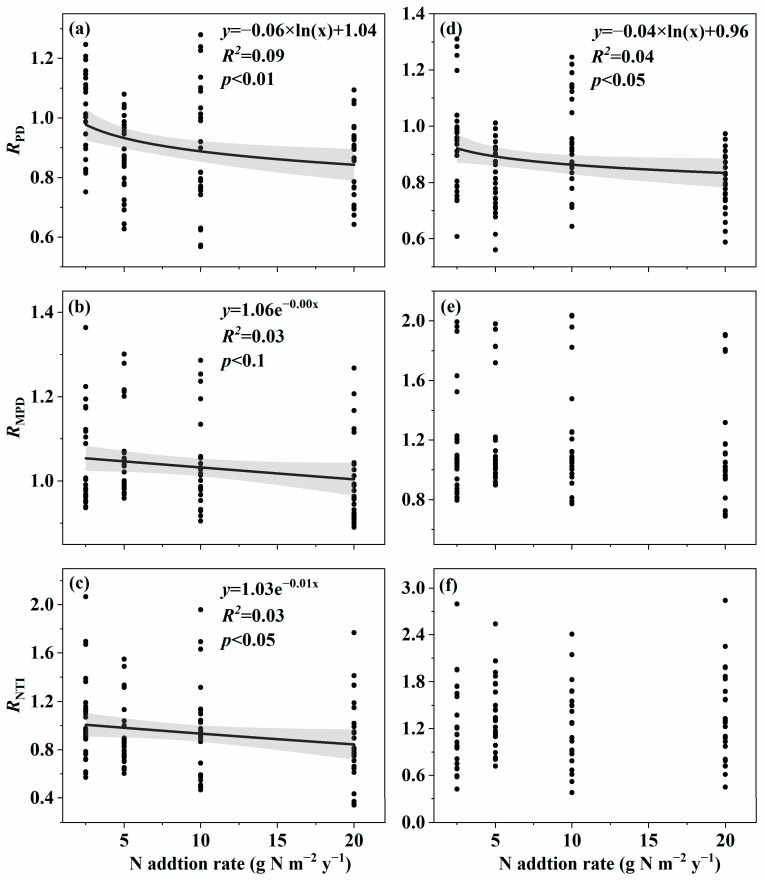
Relationships between nitrogen addition rate and the effect size of nitrogen addition on phylogenetic α-diversity of soil fungi in an alpine steppe. Surface soil: (**a**) PD; (**b**) MPD; (**c**) NTI. Subsurface soil: (**d**) PD; (**e**) MPD; (**f**) NTI.

**Figure 3 microorganisms-11-01880-f003:**
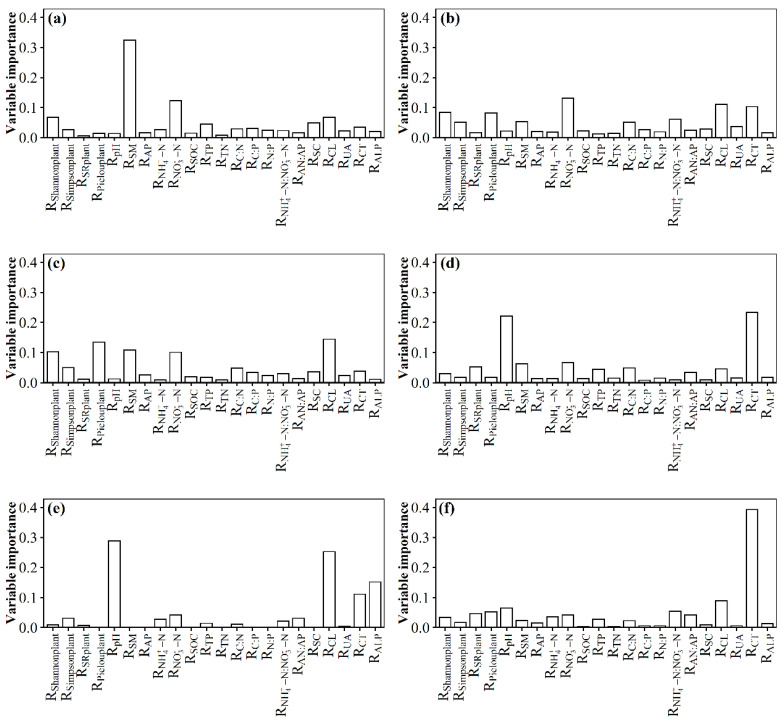
Relative contribution of observed soil and plant variables to effect size of nitrogen addition on species α-diversity of soil fungi in an alpine steppe. Surface soil: (**a**) operational taxonomic units (OTUs); (**b**) ACE; (**c**) Chao1. Subsurface soil: (**d**) OTUs; (**e**) ACE; (**f**) Chao1.

**Figure 4 microorganisms-11-01880-f004:**
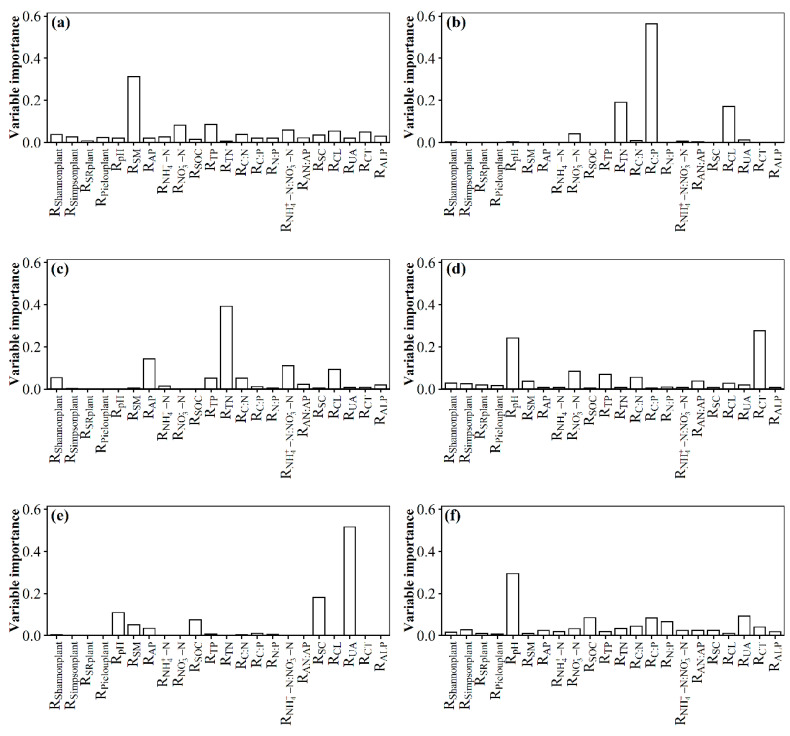
Relative contribution of observed soil and plant variables to effect size of nitrogen addition on phylogenetic α-diversity of soil fungi in an alpine steppe. Surface soil: (**a**) PD; (**b**) MPD; (**c**) NTI. Subsurface soil: (**d**) PD; (**e**) MPD; (**f**) NTI.

**Figure 5 microorganisms-11-01880-f005:**
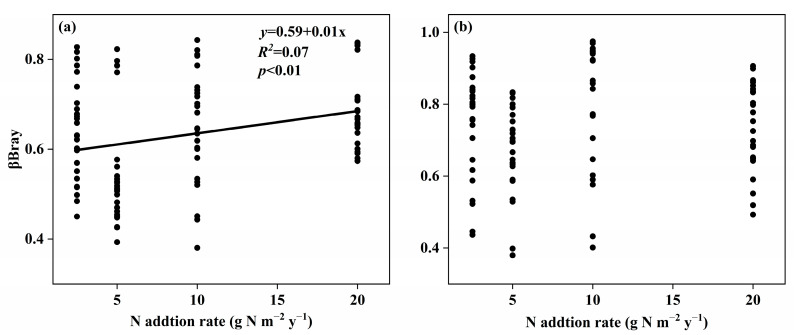
Relationships between nitrogen addition rate and the effect size of nitrogen addition on soil fungal species β-diversity in surface soil (**a**) and subsurface soil (**b**) in an alpine steppe.

**Figure 6 microorganisms-11-01880-f006:**
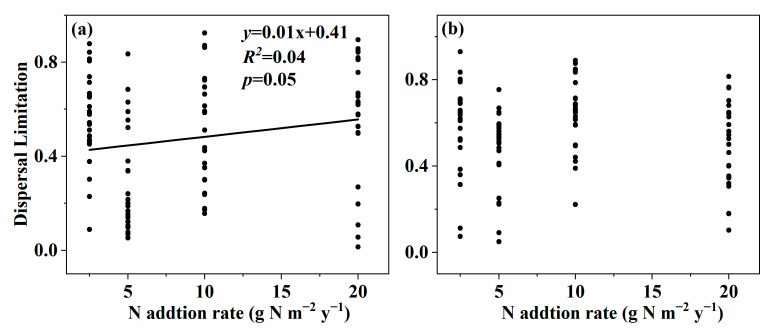
Relationships between nitrogen addition rate and the effect size of nitrogen addition on Dispersal Limitation in surface soil (**a**) and subsurface soil (**b**) in an alpine steppe.

**Table 1 microorganisms-11-01880-t001:** Significance tests of the effects of different nitrogen addition treatments on soil fungal species and phylogenetic community based on the Bray–Curtis and βMNTD dissimilarity matrix, respectively.

Species	**Surface**		**N0**	**N2.5**	**N5**	**N10**
		N2.5	0.81 (0.86)			
	Adonis2	N5	0.97 (0.51)	1.22 (0.11)		
	F (*p* value)	N10	0.96 (0.53)	0.88 (0.71)	1.27 (0.15)	
		N20	1.82 (0.02)	1.66 (0.04)	2.46 (0.01)	1.06 (0.44)
	Subsurface		N0	N2.5	N5	N10
		N2.5	0.77 (0.94)			
	Adonis2	N5	0.95 (0.62)	0.78 (0.91)		
	F (*p* value)	N10	1.17 (0.33)	0.74 (0.82)	1.55 (0.11)	
		N20	1.43 (0.06)	0.95 (0.51)	1.36 (0.09)	1.54 (0.07)
Phylogenetic	Surface		N0	N2.5	N5	N10
		N2.5	0.01 (0.79)			
	Adonis2	N5	0.20 (0.78)	0.11 (0.91)		
	F (*p* value)	N10	1.60 (0.34)	0.98 (0.52)	0.21 (0.78)	
		N20	2.71 (0.04)	1.89 (0.16)	1.09 (0.53)	0.35 (0.83)
	Subsurface		N0	N2.5	N5	N10
		N2.5	1.00 (0.55)			
	Adonis2	N5	−1.16 (0.90)	0.53 (0.77)		
	F (*p* value)	N10	3.00 (0.17)	1.06 (0.22)	3.47 (0.07)	
		N20	−0.50 (0.74)	0.20 (0.91)	1.66 (0.18)	2.53 (0.20)

**Table 2 microorganisms-11-01880-t002:** The Mantel test between soil fungal species and phylogenetic community composition and soil and plant variables based on the Bray–Curtis and βMNTD dissimilarity matrix, respectively.

Variables	Species		Phylogenetic	
	Surface	Subsurface	Surface	Subsurface
Shannon_plant_	0.07	0.14	−0.02	0.17 +
Simpson_plant_ SR_plant_	0.13	0.13	0.01	0.14
	−0.07	0.12	−0.08	0.14
Pielou_plant_	0.16	0.14	0.00	0.12
pH	−0.09	−0.17	0.00	−0.06
SM	0.09	0.14	0.28 +	0.07
AP (mg kg^−1^)	−0.04	−0.14	0.07	−0.14
NH_4_^+^-N	0.29 *	−0.11	0.20 +	−0.07
NO_3_^−^-N	0.25 **	−0.07	0.14 +	−0.10
SOC	0.19	0.24 +	−0.01	0.16
TP	0.14	0.15	0.11	0.12
TN	0.20 +	0.24 *	0.04	0.21 +
C:N	−0.14	−0.07	0.06	−0.12
C:P	0.18	0.04	0.04	−0.01
N:P	0.16	0.10	0.03	0.11
NH_4_^+^-N:NO_3_^−^-N	−0.13	−0.13	−0.13	−0.10
AN:AP	0.24 +	−0.08	0.24 +	−0.10
SC (mg g^−1^ d^−1^)	0.26 **	0.26 *	0.26 **	0.18 +
CL (mg g^−1^ 3 d^−1^)	−0.10	0.02	0.04	−0.07
UA (mg g^−1^ d^−1^)	0.06	0.34 **	0.01	0.27 +
CT (ml g^−1^ 20 min^−1^)	0.15 +	0.09	0.03	−0.05
ALP (mg g^−1^ 2 h^−1^)	−0.09	−0.08	−0.11	−0.11

Note: “+”, “*”, and “**” means *p* < 0.1, *p* < 0.05, and *p* < 0.01, respectively.

## Data Availability

Not applicable.

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
