# Peer review of "Effects of Short-Term Nitrogen Addition on Soil Fungal Community Increase with Nitrogen Addition Rate in an Alpine Steppe at the Source of Brahmaputra"

_microorganisms, 2023, doi:10.3390/microorganisms11081880_

Round 1

Reviewer 1 Report

Review of the work entitled and " Effects of short-term nitrogen addition on soil fungal community increase with nitrogen addition rate in an alpine steppe at the source of Brahmaputra " submitted to the Microorganisms journal .

In my opinion, the topic of research is good, but not groundbreaking. In turn, the methodology and quality of the conducted research is also at a good level, which enabled the researchers to obtain interesting research results.

In fact, I do not have any serious substantive comments on the manuscript. Everything was done methodically correctly. However, the authors can work on improving the readability of some Figs.

Author Response

Reviewer #1:

Reviewer #1: In my opinion, the topic of research is good, but not groundbreaking. In turn, the methodology and quality of the conducted research is also at a good level, which enabled the researchers to obtain interesting research results.

Answer: Thank you for your careful reading and helpful comments. We revised the manuscript according to them.

In fact, I do not have any serious substantive comments on the manuscript. Everything was done methodically correctly. However, the authors can work on improving the readability of some Figs.

Answer: Many thanks. We had worked on improving the readability of some Figs.

Reviewer 2 Report

Lines 103-104 – if this data is yours, it should be placed to results and the protocols of pH and carbon estimation should be given.

The part of results beginning with line 161 should be rewritten. Now it is describing the picture. All these facts I can see, but you should say something about certain differences or common observations. The definition of environmental variations like Rsm, Rtp etc. should be given in part of material and methods. So there would be no need to repeat it in a caption to figures. Also you should explain how all these parameters were counted.

Figure 3 f – it is said that the variation is 73% but if to count variation without Rct I guess the meaning will differ. Is it correct to speak about variation of all parameters if one them greatly differ from others?

It seems there are mistakes in captions of figures 5, 6S and 7S. Check this.

According to figure 1 and 2 and lines 148-151 α-diversity of soil fungal species decreased with growing nitrogen addition rate but in discussion there is another opinion (lines 287-290). And the conclusion says about negative effect of nitrogen addition on species and phylogenetic α-diversity of soil fungal community. What is correct?

There are many grammatical mistakes that make diffiuclt to understand the information. For example, lines 37, 126 are not clear.

Author Response

Reviewer #2:

Reviewer #2: (1) Lines 103-104 – if this data is yours, it should be placed to results and the protocols of pH and carbon estimation should be given.

Answer: Thank you for your comment. The results have been showed in the article of Huang et.al (2022), which we have cited in the manuscript. And we have added the assay methods of pH and soil organic carbon. Please see Line 127-128 in the change-marked manuscript.

(2) The part of results beginning with line 161 should be rewritten. Now it is describing the picture. All these facts I can see, but you should say something about certain differences or common observations. The definition of environmental variations like RSM, RTP etc. should be given in part of material and methods. So there would be no need to repeat it in a caption to figures. Also you should explain how all these parameters were counted.

Answer: Many thanks. We have revised and improved the part of results beginning with line 161. And we have given the definition of variations like ROTUs, RSM, RTP etc. in Materials and Methods. We have explained the computing method in Line 133. Please see the change-marked manuscript.

(3) Figure 3f – it is said that the variation is 73% but if to count variation without RCT I guess the meaning will differ. Is it correct to speak about variation of all parameters if one them greatly differ from others?

Answer: Yes, you are right. If we counted relative contribution of environmental variations without RCT, the result was different. We changed “The environmental variations included RCT, RCL, RpH, RNH4+-N:NO3--N, RPielouplant, RSRplant, etc., and they commonly explicated 73% variation of the RChao1 in subsurface soil with the N addition gradient (Fig. 3f).” to “The environmental variations included RCT, RCL, RpH, etc., and they commonly explicated 73% variation of the RChao1 in subsurface soil with the N addition gradient, particular the predominant factor RCT had the most relative contribution (Fig. 3f).”

(4) It seems there are mistakes in captions of figures 5, 6S and 7S. Check this.

Answer: Many thanks. We have checked the captions of figures 5, S6 and S7 and found that there were not mistakes.

(5) According to figure 1 and 2 and lines 148-151 α-diversity of soil fungal species decreased with growing nitrogen addition rate but in discussion there is another opinion (lines 287-290). And the conclusion says about negative effect of nitrogen addition on species and phylogenetic α-diversity of soil fungal community. What is correct?

Answer: Thank you for your question. The subjects of two sentences were different, whose subject was “the effect size” in Line 148-151 and “reduced magnitude” in Line 287-290. They all represented same meaning and were right.

(6) There are many grammatical mistakes that make difficult to understand the information. For example, lines 37, 126 are not clear.

Answer: Many thanks. We have revised the sentences of line 36 and 126. And we have our manuscript checked by a colleague fluent in English writing. Please see the change-marked manuscript.

Round 2

Reviewer 2 Report

The text of the article is mostly corrected. However, the part of material and methods should be added with methods of measuring of soil moisture (SM), ammonium nitrogen (NH4+-N), 118 nitrate nitrogen (NO3--N), available phosphorus (AP), soil organic carbon (SOC), total 119 phosphorus (TP) and total nitrogen (TN), and soil enzyme, including sucrase (SC), cellu-120 lase (CL), urease (UA), catalase (CT) and alkaline phosphatase (ALP). What technique did you use? What equipment was used? It has to be written. Maybe it would be good to place all this information in supplementary materials.

Again, about figures S5, S6, S7. Yes, the captions of these figures are correct, but the text that mentions these figures is somehow wrong. Line 271 –“ treatment in surface soil (Fig. S5c). Duncan multi-comparison also showed that g__Vishniaco”, but there is no c on figure S5. In supplementary figure 5 has only A and B.

Also, lines 267-268 –“There were 8 diverse clusters of 267 soil fungal species community in subsurface soil whose LDA points were >2, including 1 268 phylum, 2 orders, 4 families, and 1 genus (Fig. S6).” However, the caption of figure S6 talks about surface soil. Where is the truth? Check the text of 260-277 lines and the references of figures. 

Author Response

Reviewer #2:

Reviewer #2: (1) The text of the article is mostly corrected. However, the part of material and methods should be added with methods of measuring of soil moisture (SM), ammonium nitrogen (NH4+-N), 118 nitrate nitrogen (NO3--N), available phosphorus (AP), soil organic carbon (SOC), total 119 phosphorus (TP) and total nitrogen (TN), and soil enzyme, including sucrase (SC), cellu-120 lase (CL), urease (UA), catalase (CT) and alkaline phosphatase (ALP). What technique did you use? What equipment was used? It has to be written. Maybe it would be good to place all this information in supplementary materials.

Answer: Thank you for your comment. We have put measuring methods of soil physicochemical property in supplementary materials.

(2) Again, about figures S5, S6, S7. Yes, the captions of these figures are correct, but the text that mentions these figures is somehow wrong. Line 271 –“ treatment in surface soil (Fig. S5c). Duncan multi-comparison also showed that g__Vishniaco”, but there is no c on figure S5. In supplementary figure 5 has only A and B.

Answer: Many thanks. We have adjusted the order of the figures in supplementary materials. Now the text and the figures are matched.

(3) Also, lines 267-268 –“There were 8 diverse clusters of 267 soil fungal species community in subsurface soil whose LDA points were >2, including 1 268 phylum, 2 orders, 4 families, and 1 genus (Fig. S6).” However, the caption of figure S6 talks about surface soil. Where is the truth? Check the text of 260-277 lines and the references of figures. 

Answer: We have checked the text of 260-277 lines and the references of figures carefully. Thank you very much.
